# EGCG Inhibits Tumor Growth in Melanoma by Targeting JAK-STAT Signaling and Its Downstream PD-L1/PD-L2-PD1 Axis in Tumors and Enhancing Cytotoxic T-Cell Responses

**DOI:** 10.3390/ph14111081

**Published:** 2021-10-26

**Authors:** Dinoop Ravindran Menon, Yang Li, Takeshi Yamauchi, Douglas Grant Osborne, Prasanna Kumar Vaddi, Michael F Wempe, Zili Zhai, Mayumi Fujita

**Affiliations:** 1Department of Dermatology, University of Colorado Anschutz Medical Campus, Aurora, CO 80045, USA; dinoop.ravindranmenon@cuanschutz.edu (D.R.M.); yang.li@siat.ac.cn (Y.L.); takeshi.yamauchi@cuanschutz.edu (T.Y.); douglas.osborne@cuanschutz.edu (D.G.O.); prasanna.vaddi@cuanschutz.edu (P.K.V.); zili.zhai@cuanschutz.edu (Z.Z.); 2Department of Pharmaceutical Sciences, University of Colorado Anschutz Medical Campus, Aurora, CO 80045, USA; Michael.Wempe@cuanschutz.edu; 3Department of Veterans Affairs Medical Center, VA Eastern Colorado Health Care System, Aurora, CO 80045, USA; 4Department of Immunology & Microbiology, University of Colorado Anschutz Medical Campus, Aurora, CO 80045, USA

**Keywords:** EGCG, PD-L1, PD-L2, melanoma immunotherapy, IFN-γ signaling, JAK-STAT signaling

## Abstract

Over the last decade, therapies targeting immune checkpoints, such as programmed death-1 (PD-1), have revolutionized the field of cancer immunotherapy. However, low response rates and immune-related adverse events remain a major concern. Here, we report that epigallocatechin gallate (EGCG), the most abundant catechin in green tea, inhibits melanoma growth by modulating an immune response against tumors. In vitro experiments revealed that EGCG treatment inhibited interferon-gamma (IFN-γ)-induced PD-L1 and PD-L2 expression and JAK-STAT signaling. We confirmed that this effect was driven by inhibiting *STAT1* gene expression and STAT1 phosphorylation, thereby downregulating the PD-L1/PD-L2 transcriptional regulator IRF1 in both human and mouse melanoma cells. Animal studies revealed that the in vivo tumor-inhibitory effect of EGCG was through CD8+ T cells and that the inhibitory effect of EGCG was comparable to anti-PD-1 therapy. However, their mechanisms of action were different. Dissimilar to anti-PD-1 treatment that blocks PD-1/PD-L1 interaction, EGCG inhibited JAK/STAT signaling and PD-L1 expression in tumor cells, leading to the re-activation of T cells. In summary, we demonstrate that EGCG enhances anti-tumor immune responses by inhibiting JAK-STAT signaling in melanoma. EGCG could be used as an alternative treatment strategy to target the PD-L1/PD-L2-PD-1 axis in cancers.

## 1. Introduction

Tumor immune evasion is a hallmark of cancer cells and is mediated by many factors, including the expression of cell surface inhibitory ligands such as programmed cell death ligand 1 (PD-L1). PD-L1 interacts with the programmed cell death 1 (PD-1) receptor on T cells to induce T-cell exhaustion [1], and the monoclonal antibody therapy blocks this inhibitory interaction to re-activate T-cell-mediated tumor killing. PD-1/PD-L1-based cancer immunotherapy has, thus, resulted in remarkable clinical success for treating advanced cancers, such as melanoma [2], non-small cell lung cancer [3,4], bladder cancer [5] and others [4,6]. Another inhibitory ligand, PD-L2, is the second ligand of the PD-1 receptor to inhibit T-cell function [7] and, in consequence, both PD-L1 and PD-L2 expression is predictive of a response to anti-PD-1 therapy in cancers [8,9,10].

PD-L1 and PD-L2 expression is upregulated by multiple pathways, including NF-κB, PI3K-AKT, MAPK and JAK-STAT signaling [11,12]. Among the molecules controlling these pathways is IFN-γ, which upregulates PD-L1/PD-L2 expression through JAK-STAT signaling [13]. CD4+ T helper type 1 (Th1) cells and CD8+ cytotoxic T cells are the major cell types that produce IFN-γ in the inflammatory cellular environments [14]. Therefore, T-cell-mediated PD-L1/PD-L2 expression functions as negative feedback to balance the inflammatory responses to maintain homeostasis. However, cancer cells hijack this mechanism to attain immune evasion, making IFN-γ a double-edged sword, leading to both pro-tumorigenic and anti-tumorigenic functions [14]. For example, in the tumor microenvironment (TME), IFN-γ secreted by CD4+ and CD8+ T cells as an immune surveillance mechanism could upregulate PD-L1/PD-L2 in cancer cells, thereby contributing to tumor immune evasion via the IFN-γ-STAT1-PD-L1/L2 axis in melanoma [13]. Anti-PD-1 therapy directly inhibits this mechanism and, hence, tumors expressing PD-L1 and/or PD-L2 respond better to the treatment in multiple cancer types [15]. However, the overall response rate of PD-1/PD-L1 blockade therapy is only around 20–40%, with the highest response observed in melanoma (≈30–40%) [16,17,18]. Furthermore, the incidence of immune-related adverse events was over 64% for PD-1 inhibition and 66% for PD-L1 inhibition [19]. Hence, the improvement of this immune checkpoint blockade therapy remains a crucial area of research.

EGCG, the most abundant catechin in tea, has many biological effects, such as anti-oxidation [20], anti-tumor [21] (anti-proliferation) and anti-inflammation [22]. Previously, we reported that low doses of EGCG (up to 10 μM) inhibit melanoma cell growth by inhibiting inflammasome and IL-1beta secretion [23]. However, its effect on the cancer immune response in the TME remains unexplored. The anti-inflammatory mechanisms of EGCG involve the direct inhibition of JAK-STAT signaling [24,25] and EGCG was reported to reduce PD-L1 expression in lung cancer cells [26]. Therefore, the current study tests whether EGCG could be used as an alternative strategy to target the PD-1-PD-L1/PD-L2 signaling cascade to evoke an anti-tumor immune response in melanoma. Since many papers, including the above [24,25] and others investigating EGCG effects on tumor cell apoptosis or proliferation [27,28], used over 25 μM of EGCG, much higher than a safe dose [29], we decided to use 10 μM of EGCG for in vitro experiments and 50 or 100 mg/kg of EGCG for in vivo experiments. We demonstrate that EGCG enhanced the CD8+ T-cell-mediated tumor killing by inhibiting the tumor-intrinsic JAK-STAT-PD-L1/L2 signaling in melanoma.

## 2. Results

### 2.1. EGCG Inhibits IFN-γ-Induced PD-L1/PD-L2 Expression at the Transcriptional Level in Human Metastatic Melanoma Cells

The basal levels of PD-L1/PD-L2 are usually low in melanoma cell lines, and IFN-γ is known to induce their expression [13]. To test the effect of EGCG on IFN-γ-induced PD-L1/PD-L2 expression, we treated three human metastatic melanoma cell lines (1205Lu, A375 and HS294T) with either EGCG (10 µM), IFN-γ (10 ng/mL) or a combination of both for 24 h and analyzed the cell surface expression of PD-L1 and PD-L2. Doses ranging from 10 to 50 µM of EGCG were previously shown to inhibit IFN-γ-induced PD-L1 expression in lung cancer cells [26]. Consistent with the reported data [13], the basal levels of PD-L1 and PD-L2 were low in all three melanoma cells. EGCG further decreased their expression, but IFN-γ upregulated the cell surface expression of PD-L1 and PD-L2 in human melanoma cells. When melanoma cells were treated with EGCG and IFN-γ simultaneously, the treatment completely abolished the IFN-γ-induced upregulation of PD-L1 and PD-L2 in all three tested cell lines (Figure 1A,B).

We then tested whether EGCG affects PD-L1 and PD-L2 expression at the transcriptional level. The qRT-PCR analysis showed that EGCG downregulated PD-L1 and PD-L2 genes in 1205Lu and HS294T cells and reversed the IFN-γ-induced upregulation of PD-L1 and PD-L2 genes in all three melanoma cell lines (Figure 1C). Together, these data demonstrate that EGCG transcriptionally downregulates IFN-γ-induced PD-L1/PD-L2 expression.

### 2.2. EGCG Inhibits IFN-γ-Induced JAK/STAT Signaling in Human Metastatic Melanoma Cells

Since IFN-γ upregulates PD-L1/PD-L2 expression through JAK/STAT signaling [13], we first tested whether a JAK/STAT inhibitor ruxolitinib controlled the PD-L1/PD-L2 expression in human melanoma cells. Similar to the results shown in Figure 1A,B, the treatment of cells with ruxolitinib (10 µM) abolished IFN-γ-induced PD-L1/PD-L2 upregulation in 1205Lu and A375 cells (Appendix A). Therefore, we tested whether EGCG inhibits the JAK/STAT signaling in melanoma. EGCG downregulated the basal expression of STAT1 mRNA, but IFN-γ upregulated its expression (3–11-fold) in all three human melanoma cell lines (Figure 2A). When cells were treated with EGCG and IFN-γ, the combination reversed the IFN-γ-induced upregulation of STAT1 and further downregulated STAT1 to levels similar to EGCG-alone-treated cells in all three lines.

We then examined STAT1’s downstream target IRF1, a transcriptional regulator of PD-L1/PD-L2 genes, in 1205Lu, A375 and HS294T cells and observed similar effects (Figure 2B). While EGCG downregulated the basal expression of IRF1 mRNA, IFN-γ upregulated its expression (9–21-fold), and this upregulation was reversed by a combination of EGCG and IFN-γ treatment in all three human melanoma cell lines. Furthermore, in 1205Lu cells and A375 cells, the combination treatment with EGCG and IFN-γ downregulated IFN-γ-induced IRF1 to levels comparable to EGCG-only treated cells.

We confirmed the mRNA results by a protein analysis of STAT1 and IRF1. In agreement with mRNA changes, the IFN-γ treatment increased p-STAT1/STAT1 and IRF1 protein levels, which were abolished by adding EGCG in all three melanoma cells (Figure 2C and Appendix A). We also found that the effects of EGCG on IFN-γ-induced pSTAT1 and IRF1 were compatible with those of ruxolitinib. Together, the data confirmed that EGCG blocks JAK/STAT signaling in human melanoma cells. Since IFN-γ upregulates PD-L1/PD-L2 expression through JAK/STAT signaling [13], the EGCG-mediated downregulation of PD-L1/PD-L2 might be directed through JAK/STAT signaling inhibition.

### 2.3. EGCG Inhibits B16F10 Mouse Melanoma Growth In Vivo Comparable to Anti-PD-1 Antibody Treatment

Since EGCG downregulated PD-L1/PD-L2 expression in human metastatic melanoma cells in vitro, we speculated that EGCG treatment in mice could evoke a similar response to anti-PD-1 antibody therapy. To test this, we used mouse metastatic melanoma cell line B16F10, which has been reported to be partially responsive to anti-PD-1 antibody therapy [30], implanted in syngeneic C57BL/6 mice. PD-L2 expression has been reported to be minimal in many murine tumor models, including B16F10 cells, even after IFN-γ stimulation [10]. Therefore, we analyzed only PD-L1 expression in B16F10 melanoma cells. First, we evaluated the in vitro effect of EGCG on PD-L1 expression in B16F10 cells. In agreement with the results from human metastatic melanoma lines, EGCG treatment reversed the IFN-γ-induced PD-L1 upregulation at both protein (Figure 3A) and mRNA (Figure 3B) levels. Similarly, the EGCG treatment inhibited the IFN-γ-induced STAT1 phosphorylation and its downstream target IRF1 (Figure 3C and Appendix A).

Next, we investigated the effect of EGCG on B16F10 tumor cell growth in C57BL/6 mice and compared its effect to anti-PD-1 antibody treatment. A cup of green tea may contain 50–100 mg of EGCG [31,32], and human studies showed that up to 704 mg EGCG/day is safe when consumed through beverages [33]. A dose of 100–700 mg EGCG (1–7 cups) for humans (1.7–11.6 mg/kg) would be equivalent to that of 21–144 mg/kg for mice [34] based on Km values (https://www.fda.gov/media/72309/download. accessed on 18 October 2021) and the formula of the human equivalent dose = animal dose (kg/mg) × (mouse Km/human Km). Animal studies revealed that a 14-week treatment of mice with up to 240 mg/kg EGCG was safe [35]. Therefore, we treated mice with intraperitoneal EGCG (50 mg/kg) daily. EGCG treatment significantly suppressed the tumor growth of B16F10 cells, equivalent to anti-PD-1 antibody treatment (Figure 3D).

To test the effect of EGCG on tumor cell proliferation and apoptosis, we stained B16F10 tumor samples with Ki-67, a proliferation marker, and cleaved caspase-3, an apoptosis marker. Both EGCG and the anti-PD-1 antibody treatment reduced Ki-67-positive tumor cell numbers to almost half of untreated tumor samples (Figure 3E). However, we observed rare numbers of apoptosis (less than 3% of tumor cells) in all tumor tissues (Appendix A). The time point of harvesting tumors could explain this, as the killing of tumor cells by CD8 T cells might have occurred earlier than 14–15 days after the tumor implant. Nevertheless, the data show that both EGCG and the anti-PD-1 antibody treatment suppressed the proliferation of tumor cells.

Because the oral route of daily EGCG treatment would be more feasible in clinical practice than the intraperitoneal route, we also tested the effect of oral EGCG on melanoma tumor growth in C57BL/6 mice. Considering the reduced efficacy from oral gavage due to poor absorption and degradation, we increased the oral dose to 100 mg/kg EGCG, which has been previously used as an oral EGCG dose in mice [36]. Appendix A shows that oral the administration of 100 mg/kg EGCG achieved a similar efficacy as the 50 mg/kg EGCG intraperitoneal treatment. We did not observe any adverse effects in behavior, general health conditions, or body weights in mice treated with EGCG (Appendix A).

Together, the data confirm the in vivo efficacy of EGCG in inhibiting melanoma tumor growth, and this inhibition was comparable to anti-PD-1 antibody treatment.

### 2.4. EGCG Inhibits JAK/STAT Signaling in Tumors and Increases Granzyme Expression in CD8+ Cells in the B16F10 Tumor Microenvironment

To test whether the mechanism behind the in vivo effects of EGCG was similar to that observed in vitro, we analyzed B16F10 tumor samples for STAT1 and IRF1 expression. In agreement with the in vitro effects, treating B16F10 tumors with EGCG in vivo downregulated STAT1 and IRF1 mRNA levels (Figure 4A,B). In contrast, the anti-PD-1 antibody treatment increased the expression of STAT1 and IRF1 in the tumors. The opposing effects of EGCG and the anti-PD-1 antibody treatment on JAK-STAT signaling could be explained by their differential mechanisms of action and effects on T-cell activation and function. Figure 1, Figure 2 and Figure 3 show that EGCG directly inhibits PD-L1 expression and JAK-STAT signaling in tumor cells, which would re-activate T cells. In contrast, the anti-PD-1 antibody treatment directly blocks the PD-L1-mediated inhibitory signals to T cells. Therefore, we compared the effects of EGCG and the anti-PD-1 antibody treatment on CD8+ cells (Figure 4C–F).

Whereas EGCG did not change the number of tumor-infiltrating CD8+ cells compared to control tumors, the anti-PD-1 antibody treatment induced a considerable increase in CD8+ cells in the TME (Figure 4C) without affecting PD-L1 expression in the surrounding tumor cells (Figure 4D). On the other hand, EGCG-treated tumors showed minimal PD-L1 expression and, thus, CD8+ cells were not surrounded by PD-L1+ tumor cells. These findings suggest that, even though CD8+ cell numbers in EGCG-treated tumors were not increased, these CD8+ cells were protected from PD-L1-mediated inhibitory signals and, thus, could regain T-cell functionality.

Granzyme B is a serine protease that mediates CD8+ T-cell cytotoxicity [37]. Its expression in cytotoxic lymphocytes is considered a marker of immune activation [38,39] and predicts the immune therapy response in cancer patients [40,41,42,43]. Therefore, we analyzed granzyme B expression in CD8+ cells using tumor samples. Consistent with reported findings [43], most CD8+ cells did not express granzyme B in control B16F10 tumors (Figure 4E). However, EGCG treatment increased the number of granzyme B-expressing CD8+ cells in the TME (a 4.5-fold increase compared to untreated control tumors) (Figure 4E,F). On the other hand, the anti-PD-1 antibody treatment did not significantly change the CD8+/granzyme B double-positive cell number.

These data highlight that, although both EGCG and the anti-PD-1 antibody treatment inhibited melanoma tumor growth in mice, they possess differential mechanisms of action in tumor cells and immune cells. The findings suggest that EGCG protects CD8 cells from T-cell exhaustion by inhibiting PD-L1 expression on tumor cells. Alternatively, additional mechanisms independent of the PD-L1/PD-1 axis could affect granzyme B expression in CD8+ cells after EGCG treatment, as the anti-PD1 antibody treatment did not reverse downregulated granzyme B expression.

### 2.5. CD8+ T Cells Are Required for EGCG-Mediated B16F10 Tumor Suppression

To investigate the role of CD8+ cells in EGCG-mediated tumor inhibition, we depleted the CD8+ cell population using an anti-CD8 depleting antibody in C57BL/6 mice undergoing B16F10 implantation and EGCG treatment. The CD8+ cell depletion was confirmed by testing splenocytes from the mice for CD8 expression at the end of the experiment (Appendix A). Whereas EGCG treatment inhibited tumor growth in C57BL/6 mice, CD8+ cell depletion in EGCG-treated mice completely abolished this inhibitory effect (Figure 5). These data confirmed that CD8+ cells are required for EGCG-mediated tumor suppression in vivo.

## 3. Discussion

Immune checkpoint inhibition has recently shown great promise in the treatment of cancer patients. However, a partial response to treatment and severe adverse side effects remain a major challenge. Here, we showed that EGCG, a natural compound and the main component of green tea, inhibited mouse melanoma growth comparable to anti-PD-1 therapy. Mechanistic studies showed that EGCG targets JAK-STAT signaling and its downstream PD-L1/PD-L2 expression in tumor cells, resulting in the activation of T cells. These findings indicate that EGCG could be used as an alternative treatment strategy to induce an anti-tumor immune response in melanoma tumors.

IFN-γ secreted by immune cells activates JAK/STAT signaling [44,45] to induce PD-L1 in tumor cells and various immune cells, including macrophages, T cells, B cells and dendritic cells [46,47]. IFN-γ-induced PD-L1 expression in tumor cells drives tumor progression [48] and makes tumors dependent on PD-1/PD-L1-mediated T-cell inhibition for survival [49]. Therefore, inflamed tumors with an IFN-γ signature are more responsive to anti-PD-1 therapy. Dissimilar to anti-PD-1 therapy to block T-cell inhibition to re-activate T cells, EGCG inhibited IFN-γ-induced JAK-STAT signaling and its downstream PD-L1/PD-L2 expression in human and mouse melanoma cells; thus, protecting T-cell exhaustion and enhancing their immune responses. These differential mechanisms can explain their opposing effects on JAK-STAT signaling and PD-L1 expression in tumor samples.

While PD-L1 is constitutively expressed on many immune and non-immune cells, PD-L2 was previously considered to be expressed only on antigen-presenting cells such as macrophages and dendritic cells [12]. However, recent studies show that PD-L2 expression is not as limited as once thought and is found on tumors and endothelial cells, among others [8,12]. PD-L2 can be expressed together with PD-L1 or alone without PD-L1 in some cancer types [8]. Consistent with our findings, PD-L2 was reported to be a downstream target of IFN-γ signaling in melanoma [13] and lung cancer [50]. However, in addition to the JAK/STAT pathway, PD-L2 is also induced through STAT3/c-FOS [50] and STAT6 [51] signaling. Similarly, PD-L1 expression can also be regulated through multiple signaling pathways, including NF-κB, MAPK and PI3K/AKT [52]. Since EGCG is known to inhibit many of these pathways [53,54], the mechanisms of PD-L1/L2 downregulation by EGCG could involve inhibiting the JAK/STAT pathway and other pathways, such as STAT3, NF-κB, MAPK and PI3K/AKT. For example, we previously reported the effects of EGCG on inflammasomes and IL-1β secretion [23], which are controlled by NF-κB. Therefore, EGCG could also inhibit the crosstalk between inflammasomes and PD-1/PD-L1/2 expression.

Granzyme B is a major cytotoxic protein that drives cytotoxic T-cell-mediated tumor cell killing [37]. Granzyme B expression in post-therapy tumors has been associated with the immune therapy response in cancer [40,41,42,43]. In the anti-PD-1-sensitive CT26 murine colorectal carcinoma cell line model, granzyme B expression was upregulated 10 days after treatment and was predictive of the response [40]. However, in the B16F10 tumor model, which has also been shown to be partially responsive to anti-PD-1 antibody therapy [30], we and others [43] did not observe a significant increase in granzyme B expression after the anti-PD-1 antibody treatment, despite an increased CD8+ cell number. These data suggest that the CD8+ cells increased after the anti-PD-1 antibody treatment was dysfunctional. On the other hand, we showed that EGCG increased granzyme B expression in CD8+ cells in the TME. While we did not investigate the functionality of CD8+ cells after each treatment, our findings indicated differential mechanisms of action in EGCG and anti-PD-1 antibody therapy on CD8+ cells in the TME. The previous report showed that EGCG reduced lymphocyte proliferation in vitro, primarily in CD4+ cells and less in CD8+ cells [55]. However, in vitro effects of EGCG may be different from in vivo effects because the presence of tumor cells in the TME changes the dynamics, phenotypes and functions of immune and non-immune cells. A further investigation needs to be conducted to elucidate the direct or indirect effects of EGCG on CD8+ cells in the TME.

Furthermore, TME contains other immune cells that play critical roles in tumor immunology, such as macrophages and myeloid-derived suppressor cells (MDSCs). EGCG has been reported to negatively affect tumor-associated macrophages and M2 polarization [56,57]. EGCG has also been shown to inhibit the accumulation of MDSCs, contributing to the immune response in mouse models of breast cancer and neuroblastoma [58,59]. In the neuroblastoma model, EGCG treatment activated CD8+ T cells that were required for immune elimination, similar to our findings in this study. In addition, EGCG treatment has been reported to affect angiogenesis in the TME, which could also partially contribute to the tumor-suppressing effect of EGCG [60,61] and might explain differential effects of EGCG and the anti-PD-1 antibody treatment, reflected by Ki-67 staining shown in Figure 3E.

Together, our study demonstrated the potential use of an abundantly available green tea catechin EGCG as an immunotherapy agent for melanoma patients. We confirmed that EGCG not only downregulates PD-L1 expression in tumors, but was also similar to anti-PD1 therapy in inducing tumor regression in a mouse melanoma model. We provided evidence that CD8+ cells are required for this EGCG-mediated tumor suppression and that EGCG and the anti-PD-1 antibody treatment act differently in tumor cells and immune cells. The oral dosing of EGCG was similarly effective in reducing tumor growth in mice. These data also indicate that combining EGCG with anti-PD1 antibody may have synergistic effects in partially responsive tumors, as they have different modes of action. Future clinical and preclinical studies are required to answer these questions.

## 4. Materials and Methods

### 4.1. Chemicals and Reagents

EGCG was purchased from Tocris bioscience (Cat: 4524, Minneapolis, MN, USA) and was reconstituted in dimethyl sulfoxide (DMSO) and stored at −20 °C for in vitro experiments. For in vivo experiments, EGCG was either purchased from ENZO (Cat: ALX-270–263, Farmingdale, NY, USA) or isolated from crude green tea mixture and stored at −20 °C as powder. The isolation of EGCG was performed as follows. Two (2) grams of crude green tea mixture was dissolved in MeOH/DCM 50:50 mixture and applied to a silica gel column equilibrated with a MeOH:DCM (4:96, *v*/*v*) solvent mixture. The 4–7% solvent system was used to elute EGCG and with the 7% MeOH/DCM solvent. EGCG, was collected in 16 fractions (each 25 mL). Finally, 704 mg EGCG was obtained and was of >97% purity as confirmed via NMR and LC/MS-MS analyses. Ruxolitinib was purchased from Selleckchem (Cat: S1378, Houston, TX, USA) and was reconstituted in DMSO and stored at −20 °C.

### 4.2. Cells and Cell Culture

Human metastatic melanoma cell lines, 1205Lu, HS294T and A375, were obtained from Rockland Immunochemicals, Inc. (Pottstown, PA, USA). Mouse metastatic melanoma cell line B16F10 was obtained from ATCC (Manassas, VA, USA). Melanoma cells were grown at 37 °C in RPMI-1640 medium (Thermo Fisher Scientific, Cat: 11875093. Waltham, MA, USA) with 10% fetal bovine serum (Corning: Corning, NY, USA.) and 100 units/mL penicillin/streptomycin (Corning, Cat: 30-001-CI, Corning, NY, USA) in a 5% CO2 incubator. Cells were monitored monthly using PCR analysis [62] for mycoplasma contamination. Cell lines have been authenticated using short tandem repeat (STR) fingerprinting by the Barbara Davis Center Bioresource Core at the University of Colorado Anschutz Medical Campus.

### 4.3. In Vitro Cell Treatment

Cells were treated with 10 μM EGCG for 24 h in OptiMEM medium (Thermo Fisher Scientific, Cat: 11058021. Waltham, MA, USA) in the presence or absence of 10 ng/mL IFN-γ (BioLegend, human Cat: 570202; mouse Cat: 575302, San Diego, CA, USA). DMSO (0.1%) was used as a control for untreated cells. Ruxolitinib (10 μM) was used for the comparison of EGCG results.

### 4.4. Flow Cytometry

Cultured cells were trypsinized, washed with FACS buffer (PBS with 1% BSA) three times and stained for 30 min on ice with respective antibodies: human PD-L1 FITC (Cat: 53-5983-42) and PD-L2 APC (Cat: 17-5888-42) were from Thermo Fisher Scientific (Waltham, MA, USA), and mouse PD-L1 PE was purchased from BioLegend (Cat: 124308, San Diego, CA, USA).

Mouse spleens were collected after sacrifice and dissociated by scalpel. A cell strainer (40 µm) was used to remove the debris. Isolated splenocytes were washed with PBS and subjected to RBC lysis using RBC lysis buffer (Thermo Fisher Scientific, Cat: 00-4333-57, Waltham, MA, USA). The cells were then washed in FACS buffer and stained with respective antibodies according to the manufacturer’s protocol: anti-mouse CD3 FITC (Cat: 100204) was purchased from BioLegend (San Diego, CA, USA), and anti-mouse CD4 PE (Cat: 12-0041-82) and CD8 Alexa Flour 700 (Cat: 56-0081-80) were purchased from Thermo Fisher Scientific (Waltham, MA, USA). DAPI or PI staining was used for identifying dead cells.

The stained samples were analyzed using Gallios flow cytometer/CytoFLEX LX (Beckman Coulter, Brea, CA, USA), and the data were analyzed by Kaluza C analysis software (Beckman Coulter, Brea, CA, USA).

### 4.5. Animal Use

C57BL/6 mice (Jackson Laboratories) were used. All animal experiments were approved by the Institution Care Animal and Use Committee at the University of Colorado Anschutz Medical Campus and performed under the institutional guidelines for laboratory animals. All mice were housed with a 12 h light–dark cycle. Mice were fed with standard food and water ad libitum.

### 4.6. Mouse Tumor Experiment

Six-week-old female C57BL/6 mice were injected subcutaneously into left or right flanks with 1 × 10^5^ B16F10 mouse melanoma cells suspended in 100 μL of 25% Matrigel Matrix (BD Biosciences, San Jose, CA, USA) diluted with PBS. Tumor growth was monitored at least twice a week with an electronic digital caliper, and tumor volume was calculated according to the formula: tumor volume (mm^3^) = (length × breadth × height). On day 7 (when tumor volume reached 50 mm^3^ approximately), mice were randomly assigned into different groups and treated intraperitoneally with saline as vehicle control, EGCG (50 mg/kg) or anti-PD-1 antibody (12.5 mg/kg) every three days. Mice were sacrificed at 20 days or when the tumor reached 2000 mm^3^ in volume. Alternatively, B16F10 tumor-bearing mice on day 7 were treated with 100 mg/kg of EGCG daily by oral gavage or 50 mg/kg of EGCG daily intraperitoneally for 15 days.

CD8 depletion was performed using an anti-CD8 antibody (Cat: BE0117) from Bio X Cell (Lebanon, NH, USA). Recipient mice were injected intraperitoneally with 250 µg of depleting antibody diluted in 100 μL of PBS on days −3, 0, 3, 7, and 12 of the tumor experiment. Control mice received an equal amount of rat IgG2b isotype control (Bio X Cell, Cat: BE0090, Lebanon, NH, USA).

### 4.7. Immunofluorescent and Immunohistochemical Staining of Tissues

Formalin-fixed, paraffin-embedded B16F10 tumor tissues were cut into 4 μm sections. Paraffin was removed using xylene (Fisher Scientific, Cat: X3P, Waltham, MA, USA) and followed by ethyl alcohol (Fisher Scientific, Cat: BP2818, Waltham, MA, USA) rehydration. Antigen retrieval was performed by placing the sections in citrate buffer (pH 6.0) (Sigma-Aldrich, Cat: C9999, St. Louis, MO, USA) for 20 min at 120 °C followed by 30 min at room temperature. Endogenous peroxidase activity in the sections was quenched by immersion in 3% hydrogen peroxide (Sigma-Aldrich, Cat: H1009, St. Louis, MO, USA) for 10 min.

Tumor sections were blocked 30 min in 10% goat serum (Sigma-Aldrich, Cat: G9023, St. Louis, MO, USA) in PBS solution, and incubated overnight at 4 °C with primary antibodies: anti-CD8 (1:100; Abcam, Cat: ab203035), anti-granzyme B (2C5) (1:50; Santa Cruz, Cat: sc-8022), anti-perforin (1:200; Abcam Cat: ab16074, Waltham, MA, USA) and anti-PD-L1 (1:100; eBioscience, Cat:14-5983-82, Waltham, MA, USA). The slides were then stained by Alexa Fluor^®^ 594 labeled and Alexa Fluor^®^ 488 labeled secondary antibodies (Invitrogen, Waltham, MA, USA) for 1 h at room temperature, mounted with DAPI Fluoromount-G^®^ (SouthernBiotech, Cat: 0100-20, Birmingham, AL, USA) and analyzed using ZEISS Axioscope.

The tumor sections were also stained with a primary antibody against Ki-67 (1:1000; Abcam, Cat: 15580, Waltham, MA, USA) overnight at 4 °C. The sections stained with anti-Ki-67 were washed with PBS with Triton X-100 (PBST) and incubated with anti-rabbit HRP (1:1000; Dako, Cat: K4003, Santa Clara, CA, USA) for 1 h at room temperature. After washes with PBST, sections were developed with Vector DAB (Vector laboratories, Cat: SK-4100, Burlingame, CA, USA) for 2 min and washed in distilled water. Sections were counterstained with hematoxylin, coverslipped with Permount (Fisher Scientific, Cat: SP15, Hampton, NH, USA), and examined using Aperio ImageScope (Leica, Buffalo Grove, IL, USA).

### 4.8. Western Blot Analysis

Cells or tissue were lysed by RIPA buffer (Sigma-Aldrich, Cat: R0278, St. Louis, MO, USA) supplemented with protease inhibitors cocktail (Sigma-Aldrich, Cat: P8340A, St. Louis, MO, USA). Samples were centrifuged at 10,000× *g* for 10 min at 4 °C, and the debris was discarded. The protein concentration was determined with a DCTM Protein assay (Bio-Rad, Cat: 5000114, Hercules, CA, USA). The lysates were mixed with SDS samples buffer and incubated at 95 °C for 5 min.

Equal amounts of protein were loaded with SDS-PAGE gels and transferred onto polyvinylidene difluoride membranes (0.4 μm) at 100V for 1 h. The blots were blocked with 5% non-fat milk and incubated with primary antibody overnight at 4 °C, followed by the secondary antibody for 2 h. Blots were developed with horseradish peroxidase substrate SuperSignal™ West Femto (Thermo Fisher Scientific, Cat: 34095, Waltham, MA, USA) and analyzed by LI-COR Odyssey imaging system (LI-COR, Lincoln, NE, USA). Samples for the same experiment were run simultaneously and probed for multiple proteins. p-STAT1 (Y701) (Cat: 7649) (1:1000), STAT-1 (Cat: 14994) (1:1000) and IRF1 (Cat: 84785) (1:1000) antibodies were purchased from Cell Signaling (Danvers, MA, USA). PD-L1 antibody was purchased from Novus Biologicals (Cat: NBP1-76769, Centennial, CO, USA) (1:1000). GAPDH (Cat: PA1-987, Waltham, MA, USA) (1:4000) was purchased from Thermo Fisher Scientific.

### 4.9. RNA Extraction and Quantitative PCR Analysis

RNA was extracted by QIAGEN RNeasy kit (QIAGEN, Cat: 74136, Germantown, MD, USA) with the manufacturer’s protocol. RNA amount was quantified by an ND-1000 spectrophotometer (NanoDrop Technologies, Wilmington, DE, USA). cDNAs were reverse transcribed with total RNA (500–1000 ng per sample) using Reverse Transcription Kit (Promega, Cat: A3500, Madison, WI, USA) or iScript cDNA synthesis kit (Bio-Rad, Cat: 1708891, Hercules, CA, USA). Quantitative PCR was performed using PowerUP SYBR Green PCR Master Mix (Applied Biosystem, Cat: A25742, Waltham, MA, USA), and the samples were amplified with AriaMx Real-time PCR System (Agilent Technologies, Santa Clara, CA, USA). GAPDH or HPRT were used as in housekeeping control. The qPCR conditions were 50 °C for 2 min, 95 °C for 15 min, followed by 40 cycles of 95 °C for 15 s, 60 °C for 30 s. Primer sequences are listed in Appendix A.

### 4.10. Statistics

Student’s *t*-test was used to analyze the data from two different groups. Multiple groups were analyzed by one-way analyses of variance followed by Student–Newman–Keuls tests. The data are presented as the means ± S.D., and differences were considered significant if *p* < 0.05.

## 5. Conclusions

Together, we showed that EGCG, the most abundant green tea catechin, can be used as an immunotherapy agent for treating melanoma. EGCG inhibited PD-L1/PD-L2 expression and JAK-STAT signaling in human and mouse melanoma cells. Animal studies revealed that EGCG downregulated PD-L1 in tumor cells and induced granzyme B in CD8 cells in the TME, which was not observed by anti-PD1 antibody treatment. These data indicated distinct modes of action caused by EGCG and the anti-PD1 antibody treatment in the melanoma TME. The effect of EGCG treatment on granzyme B expression in CD8 T cells could be independent of its effect on PD-L1/PD-L2 expression in tumor cells and warrants further investigation.

## Figures and Tables

**Figure 1 pharmaceuticals-14-01081-f001:**
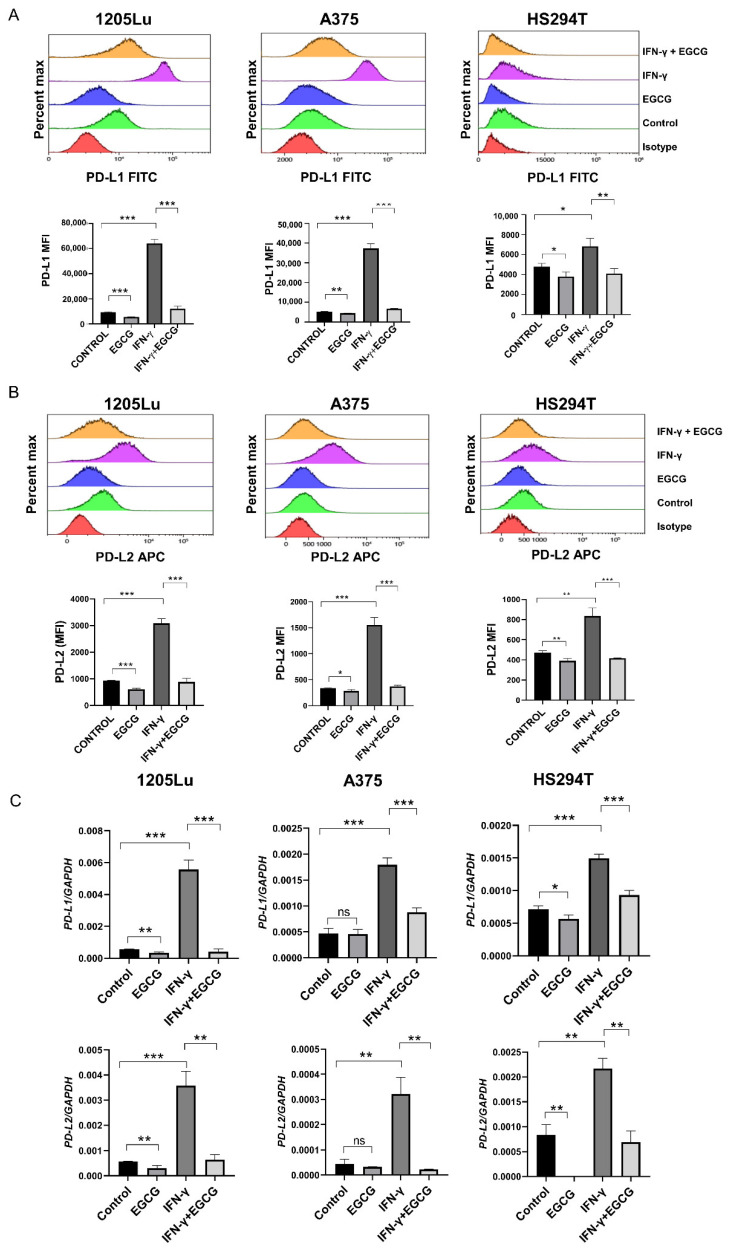
EGCG downregulated IFN-γ induced PD-L1/PD-L2 expression. (**A**,**B**) Flow cytometry data depicting the cell surface expression of PD-L1 (**A**) and PD-L2 (**B**) in 1205Lu, A375 and HS294T cells. Histogram (upper panel) and quantification of mean fluorescent intensity (MFI) (lower panel) of cells with 0.1% DMSO (control), 10 µM EGCG (EGCG), 10 ng/mL IFN-γ (IFN-γ) or a combination of IFN-γ and EGCG (IFN-γ + EGCG). (**C**) qRT-PCR analysis of *PD-L1* (upper panel) and *PD-L2* (lower panel) after treatment with 0.1% DMSO (control), 10 µM EGCG (EGCG), 10 ng/mL IFN-γ (IFN-γ) or a combination of IFN-γ and EGCG (IFN-γ + EGCG). *GAPDH* served as a control. Data are representative of 2 independent experiments and expressed as the mean ± S.D., *n* = 3 ns, *p* > 0.05; * *p* < 0.05; ** *p* < 0.01; *** *p* < 0.001.

**Figure 2 pharmaceuticals-14-01081-f002:**
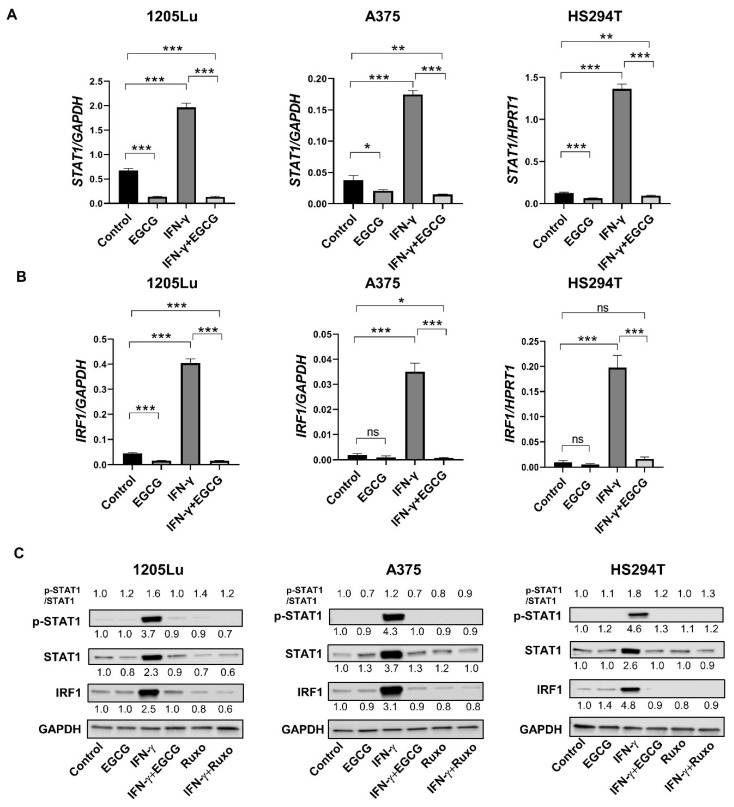
EGCG inhibits IFN-γ-induced JAK-STAT signaling. (**A**,**B**) qRT-PCR analysis of *STAT1* (**A**) and *IRF1* (**B**) in 1205Lu, A375 and HS294T cells after treatment with 0.1% DMSO (control), 10 µM EGCG (EGCG), 10 ng/mL IFN-γ (IFN-γ) or a combination of IFN-γ and EGCG (IFN-γ + EGCG). *GAPDH* served as a control. (**C**) Immunoblot analysis of p-STAT1, STAT1 and IRF1 in 1205Lu, A375 and HS294T cells treated with 0.1% DMSO (control), 10 µM EGCG (EGCG), 10 ng/mL IFN-γ (IFN-γ), a combination of IFN-γ and EGCG (IFN-γ + EGCG), 10 µM ruxolitinib (Ruxo) or a combination of IFN-γ and ruxolitinib (IFN-γ + Ruxo). GAPDH served as a control. The band densities of proteins were quantified with image J and normalized to GAPDH. p-STAT1 to STAT1 ratio was calculated and normalized to control. Data are representative of 2 independent experiments and expressed as the mean ± S.D., *n* = 3 ns, *p* > 0.05; * *p* < 0.05; ** *p* < 0.01; *** *p* < 0.001.

**Figure 3 pharmaceuticals-14-01081-f003:**
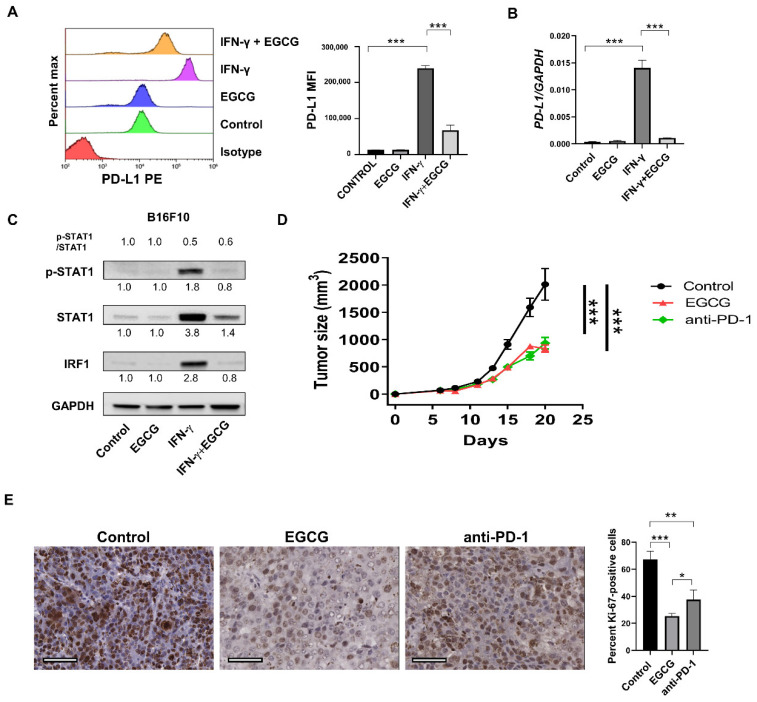
EGCG inhibited mouse melanoma growth in vivo. (**A**) Flow cytometry data depicting the cell surface expression of PD-L1 in B16F10 cells treated with 0.1% DMSO (control), 10 µM EGCG (EGCG), 10 ng/mL IFN-γ (IFN-γ) or a combination of IFN-γ and EGCG (IFN-γ + EGCG). Histogram (left panel) and quantification of mean fluorescent intensity (MFI) (right panel). (**B**) qRT-PCR analysis of *PD-L1* after treatment with 0.1% DMSO (control), 10 µM EGCG (EGCG), 10 ng/mL IFN-γ (IFN-γ) or a combination of IFN-γ and EGCG (IFN-γ + EGCG) in B16F10 cells. *GAPDH* served as a control. (**C**) Immunoblot analysis of p-STAT1, STAT1 and IRF1 in B16F10 cells treated with 0.1% DMSO (control), 10 µM EGCG (EGCG), 10 ng/mL IFN-γ (IFN-γ) or a combination of IFN-γ and EGCG (IFN-γ + EGCG). GAPDH served as a control. The band densities of proteins were quantified with image J and normalized to GAPDH. p-STAT1 to STAT1 ratio was calculated and normalized to control. (**D**) Tumor growth curve of B16F10 cells injected subcutaneously in C57BL/6 mice. Mice were treated from day 7 with saline daily intraperitoneally (control), 50 mg/kg EGCG daily intraperitoneally (EGCG), or 12.5 mg/kg anti-PD-1 antibody intraperitoneally (anti-PD-1) every 3 days. (**E**) Ki-67 staining of B16F10 tumors (derived from D) treated with saline (control), 50 mg/kg EGCG (EGCG), or 12.5 mg/kg anti-PD-1 antibody (anti-PD-1). Representative images (left panel) and quantification of positive cells (right panel). Ki-67-positive cells were counted in the whole field under a microscope and presented as % positive tumor cells. Bar = 50 μm. Data are representative of 2 independent experiments and expressed as the mean ± S.D., *n* = 3 (**A**,**B**), 10 (**D**), or 3 (**E**). * *p* < 0.05; ** *p* < 0.01; *** *p* < 0.001.

**Figure 4 pharmaceuticals-14-01081-f004:**
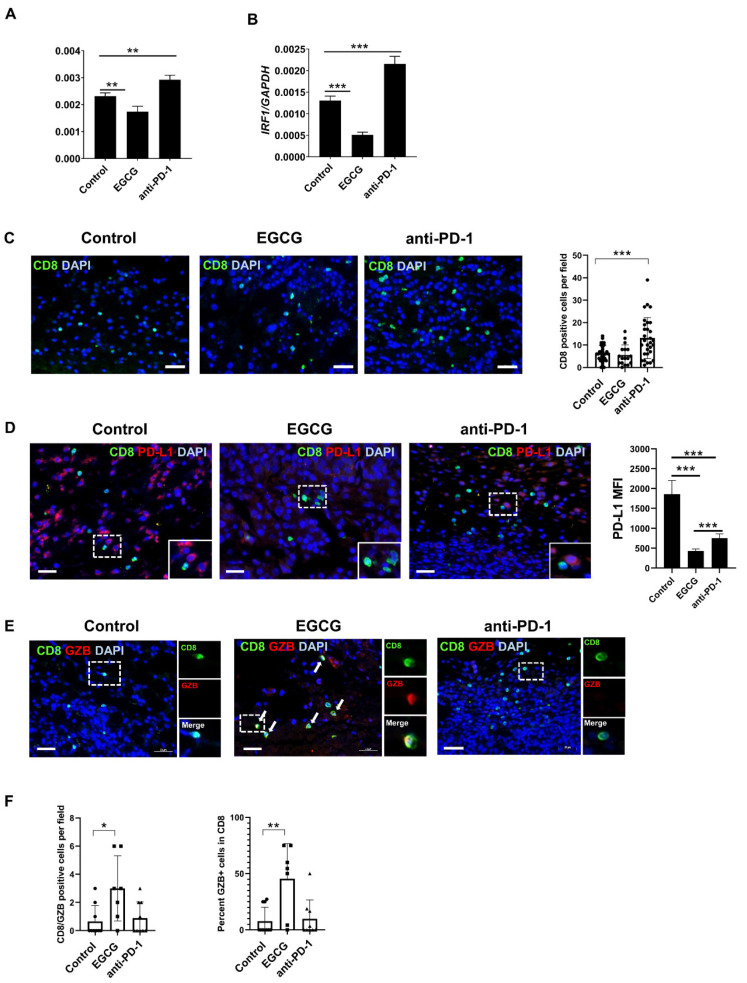
EGCG inhibited JAK-STAT signaling and increased granzyme B expression in CD8+ cells in the B16F10 tumor microenvironment. (**A**,**B**) qRT-PCR analysis of *STAT1* (**A**) and *IRF1* (**B**) in B16F10 tumors derived from Figure 3D, after the treatment with saline (control), 50 mg/kg EGCG (EGCG) or 12.5 mg/kg anti-PD-1 antibody (anti-PD-1) (*n* = 3). *GAPDH* served as a control. (**C**–**E**) Immunofluorescence study of B16F10 tumors derived from Figure 3D, treated with saline (control), 50 mg/kg EGCG (EGCG), or 12.5 mg/kg anti-PD-1 antibody (anti-PD-1). (**C**) Left, overlay images of tissue sections stained with CD8 (green) and DAPI (blue). Bar = 50 μm. Right, quantification of CD8+ cells per field (*n* ≥ 17). Each symbol represents an individual field. (**D**) Left, overlay images of tissue sections stained with CD8 (green), PD-L1 (red) and DAPI (blue). Inset: higher magnification of the dotted-line rectangular area. Right, quantification of PD-L1 expression around CD8 cells. PD-L1 mean fluorescent intensity (MFI) was calculated by measuring 90 µm^2^ area around randomly selected six CD8 cells. (**E**) Overlay images of tissue sections stained with CD8 (green), granzyme B (GZB, red) and DAPI (blue). Arrows depict CD8/GZB double-positive cells. Inset: images of the dotted-line rectangular area with CD8 only (green, upper), GZB only (red, middle) and overlay of CD8 and GZB (merged, bottom). Bar = 50 μm. (**F**) Quantification of CD8/GZB double-positive cells per field (*n* ≥ 7) (left) and % of GZB expression in CD8-positive cells (*n* ≥ 7) (right). Each symbol represents an individual field. Data are expressed as the mean ± S.D. * *p* < 0.05; ** *p* < 0.01; *** *p* < 0.001.

**Figure 5 pharmaceuticals-14-01081-f005:**
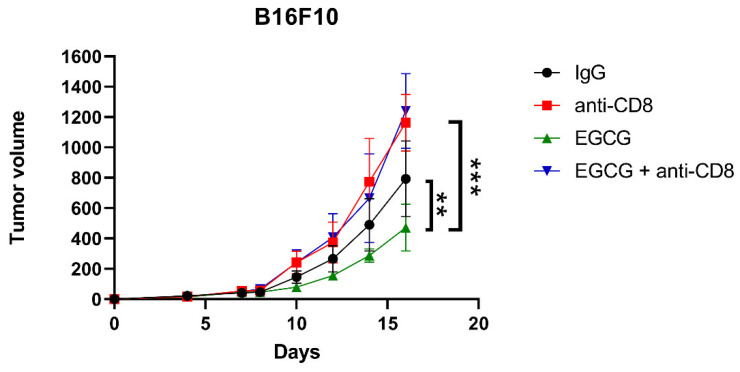
CD8+ T cells are required for EGCG-mediated tumor suppression. Tumor growth curve of B16F10 cells injected subcutaneously in C57BL/6 mice, treated with control IgG intraperitoneally (IgG), 250 μg/mouse CD8 depleting antibody intraperitoneally (anti-CD8), 100 mg/kg EGCG by oral gavage (EGCG), and a combination of EGCG and anti-CD8 antibody (EGCG + anti-CD8), *n* = 10. Data are expressed as the mean ± S.D. ** *p* < 0.01 and *** *p* < 0.001.

## Data Availability

Data is contained within the article and Appendix A.

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
