# Peer review of "EGCG Inhibits Tumor Growth in Melanoma by Targeting JAK-STAT Signaling and Its Downstream PD-L1/PD-L2-PD1 Axis in Tumors and Enhancing Cytotoxic T-Cell Responses"

_pharmaceuticals, 2021, doi:10.3390/ph14111081_

Round 1

Reviewer 1 Report

The current manuscript evaluates the effect of epigallocatechin gallate on tumor growth and immune response in different models of melanoma. The experimental approach is correct and most conclusions are supported by the evidence.

The following aspects have to be addressed or improved in order to strengthen the manuscript:

  1. Figures 1, 2, 3 and S1: according to the figure legend, only two independent experiments with n=3 were performed. “N” should refer to the number of independent experiments (i.e. biological replicates), whereby a minimum n=3 should be used.
  2. Figure 1 (FACS), according to the histograms, the signal in the isotype controls seems not significantly lower than the signal in the other groups. Could the authors be sure about the specificity of the signal in the groups control, ifn-gamma, egcg and ifn-gamma+egcg?
  3. Figures 2 and 3: please provide a quantitative analysis (optical density) for western blot data. Also, for p-STAT3 data, a ratio p-STAT3/STAT3 should be provided.

Reviewer 2 Report

In this manuscript the Authors describe the effects of EGCG in melanoma cell lines. They extensively studied the molecular mechanisms that lead to PD-L1 L2 expression through Jak-stat signaling and the cytotoxic T-cell resonse. 

Did the Authors perform a dose-dependent curve to establish the EGCG concentration to use on cell line?

Do the Authors think that EGCG could be used alone or in combination with other drugs in the treatment of melanoma? The manuscript could be improved with a "conclusion and perspective" section

Reviewer 3 Report

In their manuscript, Menon, Iee et al report that EGCG, a compound found in green tea, inhibits melanoma growth by downregulating PDL1 expression. They assessed the activity of the JAK/STAT pathway in melanoma cell lines in vitro and in the B16 mouse model in vivo by Western blotting and qPCR and found that EGCG treatment blocked the induction of the pathway by IFNg. Strikingly, EGCG reduced tumor growth in mice. This reduction was associated with increased T-cell activation as assessed by granzyme B staining, and was abolished by CD8 T-cell depletion, indicating that the EGCG anti-tumor activity at least in part depends on the reactivation of the immune system.

This study is well designed, the results are convincing and well presented. The effect of EGCG on immune modulation in cancer has been reported in lung cancer, as mentioned by the authors (Rawangkan et al, Molecules 2018), thus compromising the novelty of the present findings. Yet, the clinical potential of EGCG in melanoma might be sufficient to justify publication of this manuscript in Pharmaceuticals.

To improve their paper, the authors should consider the following minor points:

1) The authors use the term “physiological dose” of EGCG multiple times throughout their manuscript (e.g. l19, l73, l76, etc). This term is misleading and should be avoided. Indeed, EGCG is not found in the human body under physiological conditions and the authors do not report any measures of EGCG concentrations in the body (for example plasma) of individuals who drink green tea. This data would actually add value to the discussion on drug administration in mice (l145-149) as it remains unclear whether the drug concentrations used in vitro and in mouse experiments reflect the EGCG concentrations found in humans drinking green tea. Along the same line, an in vitro concentration of 10 uM should still be considered very high (l74).

2) Authors should indicate whether they noticed any toxicities or side effects associated with EGCG administration in mice. At a minimum, a measure of the weight of the animals over the course of treatment should be reported.

3) On figure 4D, could the authors quantify the PDL1 signal?

4) Adding a measure of apoptosis in tumors treated with EGCG in vivo would complement panel 3E showing cell proliferation and panel 4E showing T-cell activation.

5) In the abstract, line 27 makes it seem that the goal of anti-PD1 blockade is to induce JAK/STAT signaling in tumor cells. The authors should rephrase or delete the part of the sentence: ”and induce IFN-g-mediated JAK/STAT activation and PD-L1 upregulation in tumor cells”.

6) Line 128-129: “indicating the EGCG-mediated downregulation of PD-L1/PD-L2 is through JAK/STAT signaling inhibition”. This statement is an overinterpretation of the results and should be removed. Indeed, to prove that the inhibition of JAK/STAT signaling by EGCG is responsible for its anti-tumor effects, it would be necessary to show that forced reactivation of the JAK/STAT pathway reverses the anti-proliferative effect of EGCG.

7) Line 73-75, the authors mention that in a previous study, they showed that EGCG inhibits melanoma growth through a different mechanism. How does the present study relate to the previous one? Could the authors comment on the relative contributions of these 2 mechanisms to melanoma growth inhibition?

8) Could the authors comment more on the fact that they do not observe higher granzyme B staining upon anti-PD1 treatment in the B16 mouse model (l194-195 and discussion l250-253)?

Reviewer 4 Report

Menon et al investigated new mechanisms of EGCG, one of the components of green tea, as an anti-tumor drug by targeting immune response. EGCG reduces PD-L1 expression in tumors under inflammatory conditions by enhancing CD8+ T cell response. The findings are interesting, however, there are some concerns.

1.

Abstract: Line 19-21

we report that a physiological dose of epigallocatechin gallate (EGCG), the most abundant catechin in green tea, inhibits melanoma growth by targeting interferon-gamma (IFN-γ)-induced PD-L1 and PD-L2 expression.

In vitro, the authors do not show the EGCG effect on melanoma growth. In vivo model, no data is shown for INF- γ-induced PD-L1 and PD-L2 expression. I feel that the logic flow is not clear.

To conclude, authors should show those results in their settings.

  1. In vivo mouse model using B16, the authors do not show the data that activation of JAK/STAT and elevated PD-L1 expression is induced by IFN-γ. Authors precious paper cited as 29 do not specifically mention IFN-γ. Authors should add the data or properly cite articles.

  1. in Figure 3D and 3E, EGCG and anti-PD-1 treatment show similar inhibition levels in tumor size. However, the proliferation marker ki67 levels between EGCG and anti-PD-1 treatment are significantly different. What would be the reason?

In addition, authors should check apoptosis in mouse tumor models. If EGCG and anti-PD-1 treatment have different mechanisms in terms of tumor growth inhibition, the apoptotic rate can be different.

  1. In vivo B16 model, authors also should check PD-L1 expression in EGCG- and anti-PD-1-treated groups to prove that their in vitro findings are corresponding in vivo.

  1. Does EGCG treatment cause side effects compared to anti-PD-1 or anti-CD8 treatment? i.e. body weight loss

6.

EGCG inhibits angiogenesis in non-tumor (NATURE | VOL 398 | 1 APRIL 1999) and tumor models including B16 melanoma (Journal of Cancer Research Updates, 2014, 3, 19-29). It is possible tumor growth reduction was induced at least partially by inhibition of angiogenesis. Authors should introduce or discuss it in an introduction or discussion.

6. line 248: CT26 is a colon tumor cell line

Round 2

Reviewer 1 Report

The authors have addressed all my comments and provided an improved version of the manuscript.